# User preferences in multi-objective routes: The role of gradient visualization and personality measures

**Keisuke Otaki**[ID]*, **Takayoshi Yoshimura**

Toyota Central R&D Labs., Inc., Bunkyo-ku, Tokyo, Japan

* otaki@mosk.tytlabs.co.jp

## Abstract

Traditional pedestrian navigation systems typically prioritize the shortest or fastest routes. However, modern urban environments require multi-objective navigation that incorporates factors such as route gradient, familiarity, and individual preferences. This study investigates how presenting gradient information–either in numeric or graphical formats–affects pedestrian route choices, and how individual psychological traits, particularly the Big Five and Sensation Seeking dimensions, influence these decisions. We conducted an online survey with 91 valid participants (from an initial pool of 315), each randomly assigned to one of three groups: Control (no gradient shown), Numeric (textual gradient), or Graphical (altitude charts). Participants selected their preferred route from six route pairs, each differing in slope and distance. These pairs were generated using a multi-objective planning algorithm that optimizes both attributes. Our findings reveal three key insights. First, numeric gradient presentation led to a modest shift away from shortest-route selections (mean: 3.89 versus 4.29 in the control group), particularly for longer cases. Second, graphical gradient representations did not significantly improve decision-making over numeric formats. Third, participants with higher Sensation Seeking scores showed a significantly stronger preference for longer but gentler routes ($p < 0.01$). These results highlight the role of individual tendencies in route selection, suggesting that personalized navigation systems could be improved by incorporating user-specific psychological profiles.

## Introduction

Pedestrian navigation systems have traditionally prioritized the shortest or fastest routes. However, this practical assumption sometimes fails to capture the complexity of modern urban environments and the diverse needs of users [1,2]. These needs may include promoting health [3], enabling exploration [4], enjoying scenic aspects [5], or respecting individual route preferences [6–9]. As urban mobility becomes more personalized, a *multi-objective* approach to navigation has become increasingly essential [10–12]. Such approaches enable navigation systems to provide alternative routes that flexibly align with individual user goals and situational contexts.

**Competing interests:** The authors declare no competing interests. Although KO and TY are employed by Toyota Central R&D Labs., Inc., this research was conducted independently and was not influenced by the employer.

Multi-objective planning algorithms simultaneously optimize multiple criteria, such as distance, slope, and effort, offering a set of Pareto-optimal alternatives [13,14]. For example, a walker with physical limitations or a tourist having a heavy suitcase may prefer a slightly longer but gentler route over the shortest path. In such cases, individual preferences vary depending on context (e.g., commuting versus leisure) and personal characteristics. Despite advances in algorithmic routing, how users evaluate and select among these alternatives remains understudied.

Moreover, route choice is not only shaped by environmental factors and contexts but also by stable psychological dispositions. *Personality measures*–including the Big Five personality traits and the Sensation Seeking Scale (SSS)–capture individual differences that persist across situations [15]. The Big Five model characterizes personality along five broad dimensions (Openness, Conscientiousness, Extraversion, Agreeableness, Neuroticism), and is widely used in mobility and decision-making studies [16–18]. While these traits are known to influence travel preferences, their interaction with multi-objective route attributes remains underexplored.

Another critical factor is how route information is presented. Prior work in cartography and human-centered design has shown that visualization style–numeric values, slope charts, or visual annotations–can shape perception and decision-making [19–21]. Yet, it remains unclear whether such visual cues have a meaningful effect on pedestrian route selection in multi-objective contexts.

In this study, we aim to bridge these gaps by examining how the presentation of gradient information (numeric versus graphical) and user personality traits jointly influence pedestrian route choice. To achieve this, we employ a multi-objective algorithm to generate route pairs that differ in slope and distance. Participants recruited via crowdsourcing are shown these pairs under one of three visualization conditions (Control, Numeric, Graphical) and asked to choose their preferred route. We also assess each participant's Big Five and SSS profiles.

Based on this experimental setup, we investigate the following specific research questions:

**RQ1:** Does the explicit presentation of gradient information encourage users to select gentler, non-shortest routes?

**RQ2:** How does the mode of gradient presentation (numeric versus graphical) influence route choice?

**RQ3:** Are individual personality measures predictive of a preference for non-shortest, gentler routes?

While prior studies have examined each of these dimensions separately, this study is among the first to integrate multi-objective route generation, visualization modality, and personality traits within a unified experimental framework. By combining algorithmic optimization, visualization design, and personality profiles, we contribute to a deeper understanding of user-centered navigation. Our findings could offer practical implications for designing personalized navigation systems that consider both environmental trade-offs and individual psychological traits.

## Related research

Pedestrian route choice is a complex, interdisciplinary phenomenon influenced by environmental cues, individual traits, and cognitive processes. Tong and Bode propose a unifying framework based on four principles: information perception, information integration, responding to information, and decision-making mechanisms [22]. These principles

clarify how pedestrians perceive, process, and act upon route information in context-dependent ways. Building on this framework, we organize our literature review around three core dimensions aligned with our research questions: (1) multi-objective route generation and user preference, (2) psychological traits in mobility decisions, and (3) visualization and explainability in navigation interfaces.

## Multi-objective route planning and user preference

Pedestrian navigation systems have traditionally emphasized efficiency, typically offering the shortest or fastest routes [23–25]. However, recent studies have emphasized the importance of incorporating diverse qualitative factors, such as safety, comfort, and enjoyment, into route planning due to the known discrepancy from shortest routes in users' data [2]. Xueke and He studied the walkable and suitable routes from the perspective of environment and smart cities [26,27]. Johnson et al. proposed a classification of alternative routes into four categories: Positive, Negative, Topological, and Personalized [7]. Siriaraya et al. introduced the SWEEP taxonomy (Safety, Well-being, Effort, Exploration, Pleasure) to evaluate pedestrian route quality [1]. Feng et al. proposed comfort-enhancing navigation that optimizes routes to avoid direct sunlight [28]. Health-centered navigation has also been studied; Yang et al. tailored walking routes and food recommendations based on individual dietary preferences and health objectives [29].

To implement such quality-aware navigation, multi-objective algorithms have been developed to generate a set of candidate routes by optimizing conflicting objectives (e.g., distance and slope) [13,14]. For example, Quercia et al. used social media signals to recommend scenic or quiet paths as alternatives to the shortest route [5]. Otaki et al. proposed a resource-constrained heuristic approach for generating diverse pedestrian paths in roaming scenarios [30]. However, these studies primarily focused on the algorithmic generation of route alternatives and did not systematically evaluate how end-users perceive or select among them. More recently, several systems have extended multi-objective routing by incorporating rich urban context data and user-centric features. For instance, Routify integrates environmental layers such as air quality, noise, and green space, and offers an interactive interface with edge-level explanations and user-adjustable weights [12]. MARRS uses spatiotemporal transformer models and multitask learning to optimize route safety by predicting risks such as crime or congestion [31]. Similarly, PRoA provides accessible and green pedestrian routes by leveraging open geospatial datasets and real-time personalization on mobile devices [10]. These recent efforts demonstrate the growing attention toward context-aware and personalized routing.

Another line of research focuses not on recommending routes but on inferring user preferences from observed movement data. GPS trajectory-based approaches aim to derive desirable paths from past user behavior [2,32–34]. Such data-driven methods model route likelihood or popularity but often lack experimental control. While effective in large-scale inference, they often require careful preprocessing due to GPS noise and do not offer clear counterfactuals needed to isolate behavioral effects of specific route attributes. In contrast, our study uses algorithmically constructed route pairs in a survey environment, allowing for targeted evaluation of how users make trade-offs, such as between slope and distance, under controlled presentation formats.

## Personality traits in route or mobility decisions

Stable individual differences in personality–collectively referred to as *personality measures*–can influence travel behavior, including route selection. These measures often encompass

well-established *personality traits*, such as those defined by the Big Five model [15,35], as well as domain-specific constructs like the Sensation Seeking Scale (SSS) [36]. They reflect enduring behavioral tendencies, including novelty-seeking, risk-taking, and aversion to routine [37,38].

These personality traits have been studied across various domains, including driving [16], tourism [18], driving [17], and more recently, pedestrian mobility and wayfinding contexts. For instance, Albert et al. [16] examined whether Sensation Seeking predicts a preference for longer routes and found marginal associations in vehicular settings. Bekhor and Albert [17] demonstrated that the Thrill and Adventure Seeking (TAS) subdimension of the SSS influenced route choice under conditions of time uncertainty. These findings suggest that personality can affect route decisions involving physical or temporal effort, supporting the inclusion of psychological traits in mobility models.

Other recent studies have extended this perspective by linking user state and instruction demand in real-world wayfinding settings. Alinaghi et al. found that instruction request frequency was influenced by both internal cognitive state and environmental complexity, suggesting that some individuals may be more prone to uncertainty or distraction [39]. Such tendencies may correlate with traits, such as Neuroticism or SSS. Collaborative navigation research has also highlighted how interpersonal factors shape decision-making. Bae and Montello showed that role adoption (leader/follower), negotiation behavior, and route improvisation vary systematically between dyads, reflecting individual and social cognitive differences [40] These findings underscore the value of accounting for psychological variation, even within real-world pedestrian settings.

Despite these developments, few studies have examined how personality traits modulate route selection in *multi-objective* pedestrian navigation–where trade-offs between effort and efficiency are made explicit. Moreover, standard modeling approaches such as discrete choice analysis [41–43] face limitations due to sample size or user burden [44,45], particularly when evaluating subtle individual-level effects. By using controlled, pairwise comparisons of algorithmically generated route alternatives, our study offers a tractable approach for investigating how personality traits influence pedestrian route preferences under well-defined trade-off conditions, in contrast to prior observational studies that rely on retrospective trajectory analysis.

## Visualization and explainability in navigation interfaces

The presentation format of route information plays a critical role in how users interpret and evaluate navigation options. In pedestrian navigation, route annotations–whether textual, graphical, or symbolic–can influence perception of difficulty, effort, and preference. This section reviews key findings related to map design, visual cognition, and the effectiveness of explanations in route decision contexts.

Foundational work in cartography introduced the concept of visual variables, such as line width, color, and orientation, that shape how spatial information is perceived [20]. Building on this, Fuest et al. [19,21] demonstrated that slope charts can enhance understanding of elevation effort, although their effectiveness depends on visual clarity and layout parameters. Another line of studies has empirically tested how different elevation visualizations affect route planning. In bicycle map tasks, Brügger et al. [46] found that while users preferred elevation profiles, they performed more accurately with color-coded arrows. This discrepancy between preference and performance highlights the importance of evidence-based visual design.

Beyond route-specific visual encodings, overall map readability is affected by cognitive load. Cheng [47] conducted a neurocognitive study showing that spatial learning was optimal when a moderate number of landmarks were displayed; too many visual elements increased mental workload. These results emphasize the need to calibrate visual richness to user capacity, especially in mobile or space-limited interfaces. Visualization effectiveness is also moderated by individual differences. Münzer and Stahl [48] compared map-based and egocentric visualizations (static versus animated) in indoor wayfinding. Animations improved performance, particularly for participants with lower spatial ability or a self-reported poor sense of direction. Although interaction effects were not statistically significant, the study suggests that visualization style interacts with user cognition.

Mobile navigation platforms further complicate display design due to hardware constraints. Gartner et al. and Perebner et al. reported that small-screen interfaces (e.g., smartwatches) reduce spatial comprehension and increase user burden [49,50]. Their findings are especially relevant when presenting complex attributes, such as elevation, slope, or accessibility. Complementing spatial design research, the field of explainable recommendation has emphasized the role of transparency in influencing user trust and decision-making [51–54]. Studies show that users are more likely to follow suggestions–especially non-shortest paths–when the rationale is explicitly communicated.

Our study builds on these insights by comparing numeric and graphical annotations of route gradient, alongside a no-information control condition. This allows us to isolate the effect of annotation modality on route preferences in a multi-objective setting. Furthermore, by combining annotation formats with personality profiles, we examine how visualization interacts with individual traits in shaping user decisions.

## Materials and methods

This section describes our study design, including (1) the generation of route alternatives using multi-objective optimization, (2) the overall system pipeline from data preprocessing to user interface, (3) the design of visualization conditions, and (4) the procedures for collecting personality and choice data.

### System overview

Fig 1 outlines the complete pipeline of our experimental system: building problem sets and collecting participants for experiments. We begin by extracting pedestrian road networks from OpenStreetMap [55] and augmenting them with elevation data obtained via the Google Maps Platform Elevation API. Based on pre-defined origin–destination pairs, we compute candidate routes using the BOA* algorithm [13], which supports Pareto-optimal search across multiple objectives. In our case, the two objectives are travel distance and cumulative ascent (positive slope).

From the resulting Pareto-optimal set obtained by BOA*, we select a fixed shortest route (route A) and an alternative route (route B) that prioritizes slope reduction while limiting detour distance. Specifically, route B is constrained to fall within a detour threshold of either +10% or +30% in distance. These instances are referred to as *short-distance* and *long-distance* cases, respectively. The route pairs are then visualized in one of three annotation styles and presented to participants in a web-based interface. Each participant completes a series of pairwise route choice tasks and responds to personality questionnaires.

This design allows us to examine how participants balance trade-offs between route distance and slope in controlled, interpretable choice settings. See our S1 Appendix. Method details for details of this procedure and our experimental field.

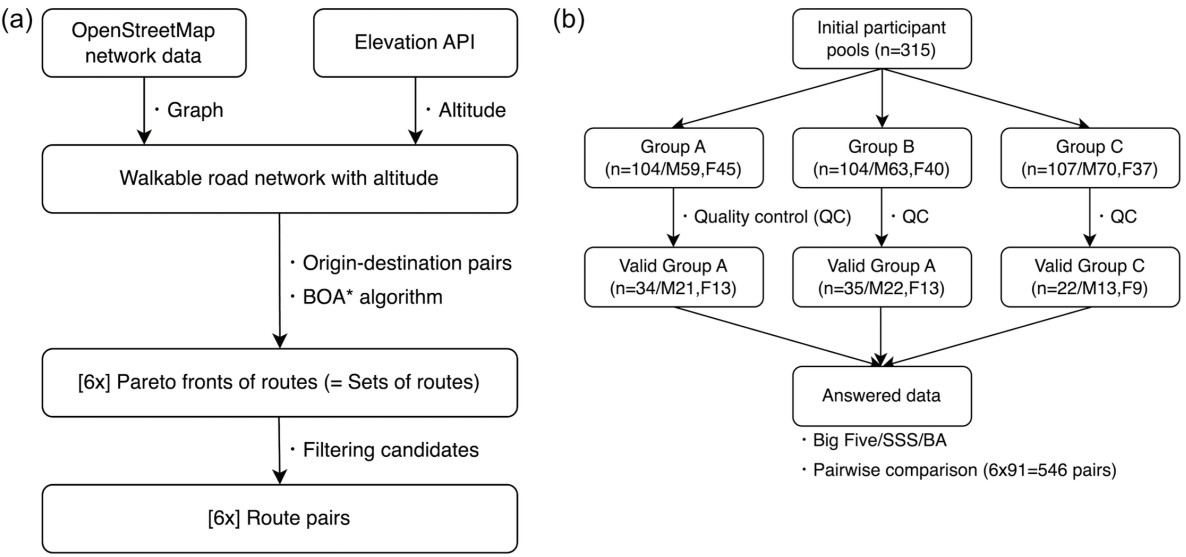

**Fig 1. Overview of the experimental design and data processing.** (a): Route generation process: multi-objective optimization using Open-StreetMap (OSM) data and elevation API to generate six route pairs. (b): Participant assignment and validation: recruitment of 315 users, group assignment, and final valid participant counts for each group.

## Multi-objective route generation

To generate alternative pedestrian routes, we apply a multi-objective planning approach based on BOA* [13], which enables efficient exploration of the Pareto front. For each origin-destination pair, we extract a subgraph from OpenStreetMap with elevation information and compute routes that simultaneously minimize total distance and cumulative ascent. Details of preprocessing and implementation are provided in S1 Appendix. Method details.

From the set of Pareto-optimal solutions, we first identify the shortest route (route A). We then scan for a contrasting route (route B) that achieves lower slope while staying within a relative detour threshold from route A (either +10% or +30%). These values were chosen based on prior studies that report commonly accepted route deviation tolerances in real-world navigation scenarios. For example, deviations of +5%–+20% are often observed in pedestrian GPS trajectories [56–58], while simple experimental setups have used 20%–30% deviations to induce observable trade-offs [6,59]. Given the complexity of urban pedestrian networks with stairs, slopes, and detours, we adopted +10% and +30% to construct balanced trade-off scenarios.

Although only two routes are presented at a time, both are selected from a multi-objective search space and thus preserve the structure of the underlying trade-offs. This pairwise format enables cognitively tractable preference elicitation while maintaining the relevance of multi-objective optimization. Such designs are consistent with prior work in route choice modeling [17,44], where pairwise judgments are favored for realism and user burden reduction.

## Participant grouping

To evaluate how route visualization influences decision-making, participants were randomly assigned to one of three visualization conditions:

- **Group A (Control, Fig 2(a))**: Route maps without any explicit gradient information.

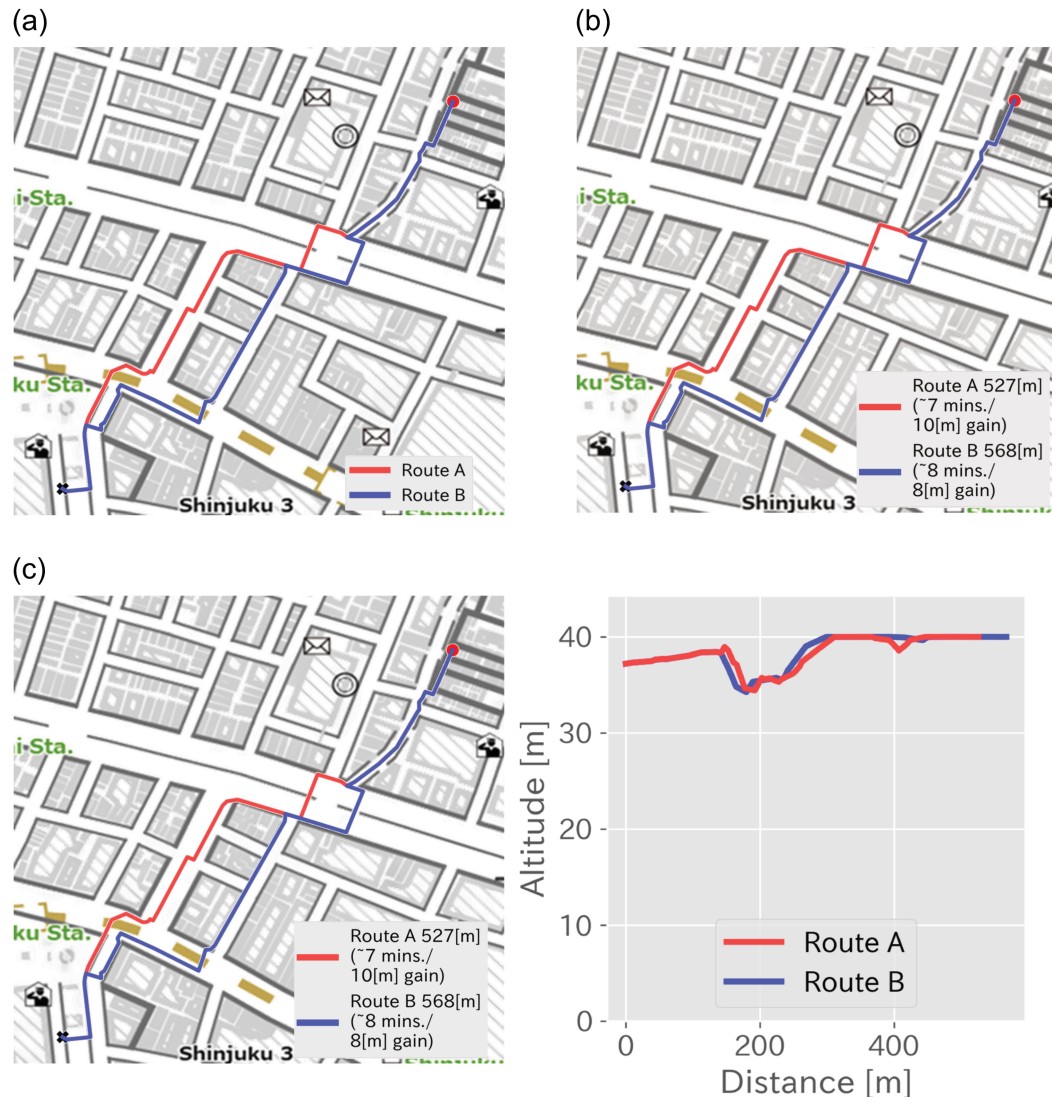

**Fig 2. Three types of visualization methods, corresponding to groups.** (a): Route visualization without annotations for Group A. (b): Routes explained with texts for Group B. (c): Routes with attached altitude history for Group C. These maps illustrate how our online survey questions are presented. The maps are based on Digital Japan Basic Maps with English labels, published by Geospatial Information Authority of Japan, whose license PDL 1.0 (Public Data License, Version 1.0), compatible with CC BY 4.0 [60–63].

- **Group B (Numeric, Fig 2(b))**: Gradient information displayed as total elevation gain in meters (e.g., "+8 m"), added as text.
- **Group C (Graphical, Fig 2(c))**: Gradient information visualized as a line chart showing altitude profiles along each route.

The assignment was implemented via randomized survey branching logic in the online platform. Participants were unaware of alternative visualization styles assigned to other groups.

**Table 1. Questions for Big Five personality traits [35].**

| Index | Category | Question: I see myself as |
|---|---|---|
| 1 | Ex. | Extraverted, enthusiastic. |
| 2 | Agr. [R] | Critical, quarrelsome. |
| 3 | Con. | Dependable, self-disciplined. |
| 4 | Neu. | Anxious, easily upset. |
| 5 | Op. | Open to new experiences, complex. |
| 6 | Ex. [R] | Reserved, quiet. |
| 7 | Agr. | Sympathetic, warm. |
| 8 | Con. [R] | Disorganized, careless. |
| 9 | Neu. [R] | Calm, emotionally stable. |
| 10 | Op. [R] | Conventional, uncreative. |

Note: Ex. = Extraversion, Agr. = Agreeableness, Con. = Conscientiousness, Neu. = Neuroticism, Op. = Openness.

Each participant was asked to complete six pairwise route selection tasks under their assigned condition. In each task, participants viewed two alternative routes (A and B), presented on a map with group-specific annotations, and were asked to choose the route they preferred. Following the choice tasks, participants completed personality questionnaires and demographic information forms.

This between-subjects design enabled us to isolate the effect of annotation style on route preference while avoiding carryover or learning effects that may arise in within-subject designs.

## Personality measures

To investigate how psychological traits influence route choice, we collected three sets of personality measures through self-report questionnaires. These include (1) the Big Five personality traits, (2) the Sensation Seeking Scale (SSS), and (3) the Boredom-Alleviation subdimension from the novelty-seeking scale. The questionnaire sheet can be checked in S2 Appendix. Questionnaire sheet. Each scale captures a distinct dimension of user preference tendencies relevant to mobility and decision-making.

**Big Five personality traits.** The Big Five is a widely used framework that characterizes personality across five dimensions: Openness (Op.), Conscientiousness (Con.), Extraversion (Ex.), Agreeableness (Agr.), and Neuroticism (Neu.) [15,35]. We used a 10-item short form (TIPI-J), validated for Japanese participants, in which each item is rated on a 7-point Likert scale from "Disagree strongly" to "Agree strongly" [35]. The 10 items and their corresponding categories are shown in Table 1. Half of the items are positive, and the other half are negative, denoted by [R]. For each category (e.g., Ex), the response scores are summed, where the scores of negative items are reversed.

**Sensation Seeking Scale (SSS).** The SSS captures individual tendencies toward novelty, risk, and stimulation [36]. We employed the brief 8-item version validated by Hoyle et al. [64], using 5-point Likert responses. It includes four subdimensions: Experience Seeking (ES), Boredom Susceptibility (BS), Thrill and Adventure Seeking (TAS), and Disinhibition (Dis). Table 2 lists the items used in this study, adapted to Japanese participants. For each category (e.g., BS), the response scores are summed.

**Boredom-alleviation from the novelty-seeking scale.** To capture motivation toward routine-breaking behaviors, we also included the Boredom-Alleviation (BA) subdimension

**Table 2. Items for the brief Sensation Seeking Scale (SSS) [64].**

| Index | Category | Question: Answer on a 5-point Likert scale |
|---|---|---|
| 1 | ES | I would like to explore strange places. |
| 2 | BS | I get restless when I spend too much time at home. |
| 3 | TAS | I like to do frightening things. |
| 4 | Dis | I like wild parties. |
| 5 | ES | I would like to take off on a trip with no pre-planned routes or timetables. |
| 6 | BS | I prefer friends who are excitingly unpredictable. |
| 7 | TAS | I would like to try bungee jumping. |
| 8 | Dis | I would love to have new and exciting experiences, even if they are illegal. |

Note: ES = Experience Seeking, BS = Boredom Susceptibility, TAS = Thrill and Adventure Seeking, Dis = Disinhibition.

from the novelty-seeking inventory [37]. This scale consists of 3 items, rated on a 5-point Likert scale, and is independent of the SSS. Table 3 lists the items used. The response scores are summed.

## Online survey procedure

Data were collected via an online survey implemented using JotForm [65] and distributed through CrowdWorks [66], a Japanese crowdsourcing platform. Participants were recruited from the Kanto region of Japan and assigned to one of the three visualization groups.

Each participant completed the following steps:

1. Agreed to the study's consent form.
2. Completed six route choice tasks, each presenting a pair of annotated routes (A and B).
3. Provided optional free-text comments for each task.
4. Submitted demographic information.
5. Completed the personality questionnaires described above.

To ensure data quality, two dummy tasks were embedded–one in the short-route set (approximately 500 meters) and one in the long-route set (approximately 1000 meters)–in which the correct choice was instructed. Participants who failed either dummy task were excluded from analysis. Participants took approximately 10–30 minutes to complete the survey and received a reward of 200–300 JPY.

Our experimental protocol was approved by our institution's ethics committee [Toyota Central R&D Labs., Inc., Ethics Committee] (approval ID: 23A-29). Our experiment was planned on Nov. 2023. The recruitment and survey were conducted between January 13 and May 18, 2024. We expect the collection of valid patricipants up to 100. All participants were informed about the purpose and structure of the study and provided written informed consent by agreeing to a consent form at the beginning of the online survey. Only participants who gave consent were allowed to proceed, and they could opt out at any time without any

**Table 3. Items for the Boredom-Alleviation (BA) scale [37].**

| Index | Question: Answer on a 5-point Likert scale |
|---|---|
| 1 | I want to travel to relieve boredom. |
| 2 | I have to go on vacation from time to time to avoid getting into a rut. |
| 3 | I like to travel because the same routine work bores me. |

penalty. The study was conducted entirely online using a desktop interface and map images rendered in Japanese, corresponding to real-world locations in Shinjuku, Tokyo.

## Results

The collected data were analyzed, and our findings were discussed. The raw route-choice data underlying our analyses are provided in Supporting Information S3 File. Dataset and script, including both the original voting records and pre-processed data used for generating the figures presented in this section.

### Participant statistics and valid responses

A total of 315 individuals accessed the survey. After applying quality control filters based on dummy questions, we obtained valid responses from 91 participants (Control: 34, Numeric: 35, Graphical: 22). Table 4 summarizes the demographic information. Fig 3(a) and 3(b) show the histograms of the entered ages and gender of all participants and valid participants. Most participants were in their 30s or 40s, reflecting an active adult demographic. For each valid participant who passed the dummy questions, we collected the entered route-level preference data on six pairs of routes. Finally, we obtained $(34 + 35 + 22) \times 6 = 91 \times 6 = 546$ pairwise comparison data, where 91 was the total number of valid participants.

### Route choice behavior across groups

Fig 4(a) summarizes the average number of shortest route selections (out of 6) per group. Participants in the control group (A) selected the shortest route most frequently (mean = 4.29), while participants in the numeric group (B) selected it less often (mean = 3.89). This suggests

**Table 4. Participant group assignment and demographics.**

| Group | Valid participants | Male | Female |
|---|---|---|---|
| Group A: Control | 34 | 21 | 13 |
| Group B: Numeric | 35 | 22 | 13 |
| Group C: Graphical | 22 | 13 | 9 |
| Total | 91 | 56 | 35 |

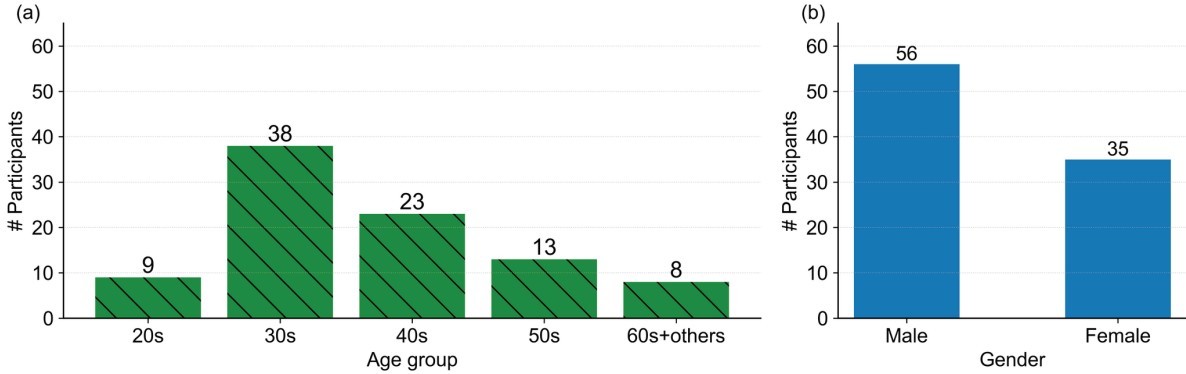

**Fig 3. Valid participants statistics:** *x*-axes represent age groups in (a) and Gender in (b), and *y*-axes show numbers of participants (denoted by # Participants). The values on both bar plots represent the corresponding numbers of participants.

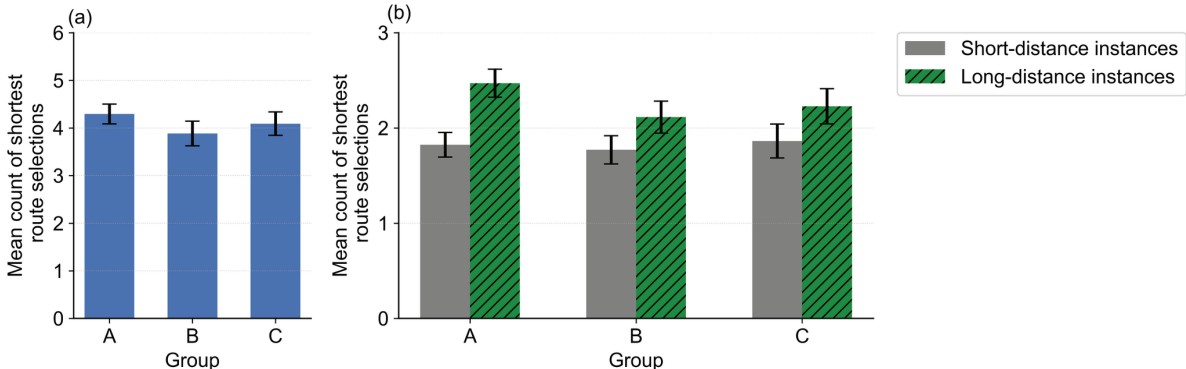

**Fig 4. Mean counts of shortest route selections: (a) counts per group and (b) counts per group and distance.** The error bars denote the standard errors of the mean.

that numerical annotations may modestly encourage selection of the gentler route. The graphical group (C) showed a similar pattern to group A (mean = 4.09), indicating that graphical annotations did not yield additional benefits over numeric values.

To examine the role of route length, we analyzed choices separately for short-distance (+10%) and long-distance (+30%) cases in Fig 4(b). In short-distance cases, the distribution of shortest route choices was similar across groups. However, in long-distance cases, participants in the numeric group were more likely to select the longer but gentler route compared to the control.

These findings partially support RQ1 and RQ2: explicit gradient information affects route selection, especially when the cost of detour is substantial. However, graphical annotation does not provide a significant additional advantage over numeric display.

## Impact of personality measures

To evaluate how individual psychological traits relate to route selection (RQ3), we first clustered participants into two groups based on their route choice patterns: (1) the **GENTLER_G** group, who more frequently selected routes with gentler gradients, and (2) the **SHORTER_G** group, who preferred the shortest routes. Fig 5 shows the distribution of shortest-route selections across participants.

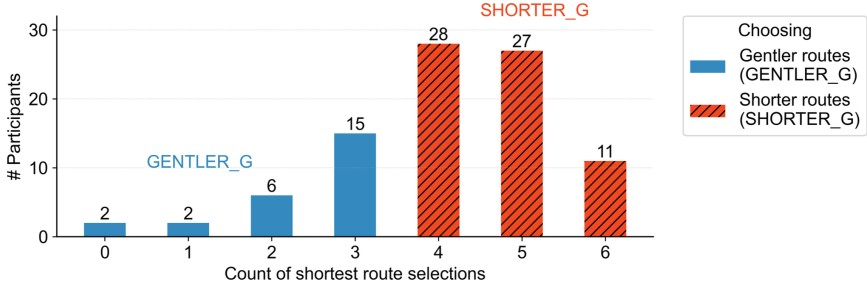

**Fig 5. GENTLER_G (blue) and SHORTER_G (red) clusters, representing participants who tended to choose gentler routes and shorter routes, respectively.** The values on both bar plots represent the corresponding numbers of participants.

Personality trait scores were compared between these two groups. Fig 6 presents the distributions for the Big Five, Sensation Seeking Scale (SSS), and Boredom-Alleviation (BA). Significant differences were found across all four SSS subdimensions: Experience Seeking (ES), Boredom Susceptibility (BS), Thrill and Adventure Seeking (TAS), and Disinhibition (Dis) ($*: p < 0.05$ or $**: p < 0.01$, Mann-Whitney U-test). The detailed plot of the BA subdimension is also illustrated in Fig 7. No significant group differences were observed in Big Five traits or the BA subdimension.

To further examine this effect, participants were clustered into two new groups based on their overall SSS scores: **LOW_SSS** (safety-oriented) and **HIGH_SSS** (challenge-oriented). As shown in Fig 8, these groups were then compared by distance condition.

Fig 9 shows that HIGH_SSS participants chose gentler routes more frequently than LOW_SSS participants ($p < 0.001$), especially in long-distance scenarios. By contrast, age and

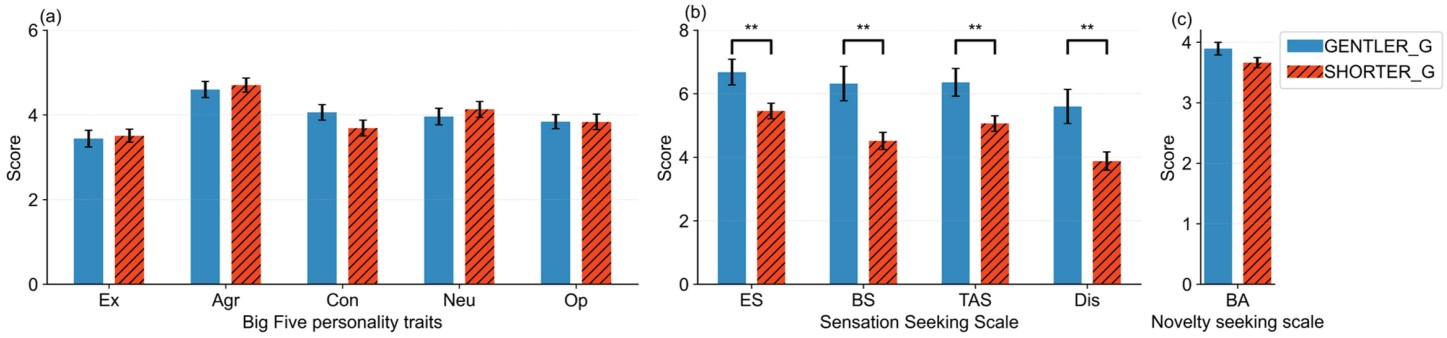

**Fig 6. Scores for each factor for GENTLER_G and SHORTER_G in (a) Big Five, (b) SSS, and (c) BA.** The error bars denote standard deviations.

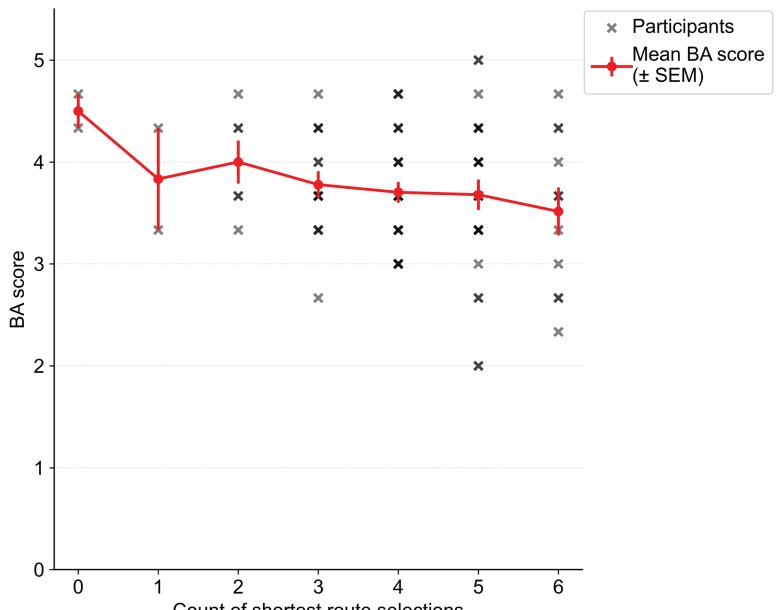

**Fig 7. Detailed plots of the BA (shortest route selections in *x* and scores in *y*).** The points represent individuals, and the red lines show the mean values per *x*.

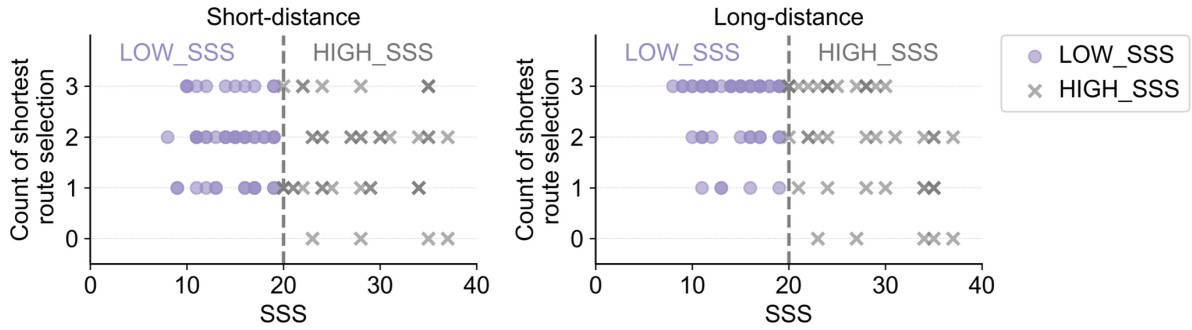

**Fig 8. Grouping participants into LOW_SSS (circle) and HIGH_SSS (cross) in each distance cluster.**

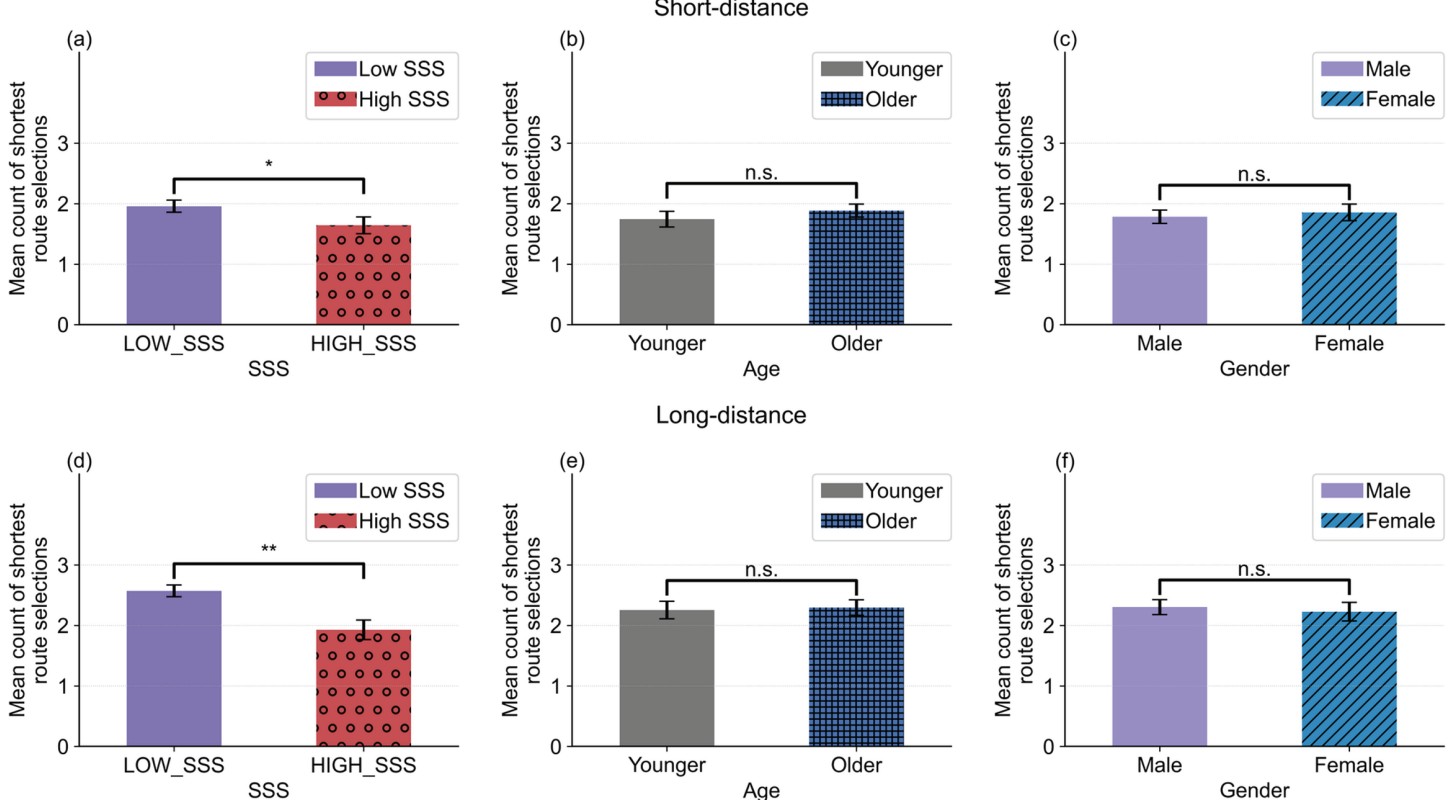

**Fig 9. Comparison of the mean counts of shortest route selections between participants with SSS ((a) and (d)), Age ((b) and (e)), and Gender ((c) and (f)) clusters, corresponding to x-axis.** Instances are grouped in y-axis (Short-distance in (a)–(c) and Long-distance in (d)-(f)). Error bars denote the standard errors of the mean. In each subfigure, the legend represents the two groups compared with different visualizations.

gender did not show significant associations with route choice, suggesting that the SSS trait uniquely predicts detour preferences.

To quantify the magnitude of these psychological differences, we computed Cohen's $d$ effect sizes between the GENTLER_G and SHORTER_G groups for each subdimension of SSS. The results revealed medium to large effects across all four subdimensions: Experience Seeking (ES; $d = 0.61$, 95% CI [0.30, 2.15]), Boredom Susceptibility (BS; $d = 0.77$, 95% CI

[0.62, 2.99]), Thrill and Adventure Seeking (TAS; $d = 0.64$, 95% CI [0.32, 2.28]), and Disinhibition (Dis; $d = 0.70$, 95% CI [0.53, 2.91]). These results underscore the robust association between higher SSS traits and a greater preference for gentler, non-shortest routes. In particular, the BS and Dis subdimensions showed the largest effects, suggesting that those who avoid routine and favor uninhibited decisions are more inclined to deviate from default (i.e., shortest) paths. In addition, the overall SSS scores were significantly higher in the GENTLER_G group than in the SHORTER_G group ($d = 0.80$, 95% CI [2.26, 9.84], $p < 0.001$), reinforcing the global association between sensation seeking and route preference behavior.

To better interpret the role of individual SSS subdimensions in route preferences, we consider the behavioral meanings of each trait and how they relate to mobility decisions. Participants scoring high in ES are generally open to novel environments and spontaneous exploration. This aligns with our finding that such individuals were more willing to accept longer but non-standard routes, consistent with prior work in tourism behavior [18].

BS reflects discomfort with routine and a desire for stimulation. This trait may explain why participants with higher BS scores avoided the shortest (i.e., routine-like) routes, favoring alternatives even with added effort. Lee and Crompton [37] noted similar tendencies in novelty-seeking tourists who deliberately avoid repetitive paths.

TAS, as used in mobility contexts [17], typically captures a willingness to take risks or engage in physical challenge. While walking steeper routes might not be "risky" in a strict sense, the willingness to diverge from efficient norms could still reflect a thrill-oriented mindset.

Dis describes a general readiness to act on impulse and ignore convention. In the context of our task, higher Dis scores might predict openness to choose non-default options without being constrained by normative assumptions (e.g., shortest path = best).

Taken together, these findings suggest that route preferences are shaped not only by spatial attributes but also by stable personality dispositions–each affecting route interpretation in distinct ways. This insight reinforces the potential of integrating personality-based user modeling into future pedestrian navigation systems.

In summary, participants with higher sensation seeking scores (especially in TAS, ES, BS, and Dis) showed a stronger preference for longer but gentler routes. This relationship was amplified for routes with longer detours. These findings highlight the predictive power of SSS traits and suggest that integrating psychological profiles into navigation systems may support better user-specific route recommendations.

## Qualitative insights from participant comments

When collecting our datasets, we also collected free comments from the participants. Here, we reviewed the comments and formed clusters to gain insights from the participants and aid the development of non-shortest route recommender systems.

**Regional effects.** A participant commented "*I am familiar with the targeting area. Therefore, I do not want to select detour routes even if the detour is short.*" In contrast, another participant wrote "*I would prefer a gentle route as I have a leg problem.*" Some participants mentioned that "*Routes along with more safe area are better.*" For these users, recommending routes using additional information (e.g., safety and sidewalk widths) could be effective.

**Additive values.** Another participant stated, "*I would like to take longer routes for exercise.*" In addition, a few participants commented, "*I want to walk streets with seeing trains.*" Based on these comments, we consider providing additional values rather than the minimum distance would effectively enhance the users' walking experiences, and including

information about scenic imagery, areas, and point-of-interests (POIs), e.g., shops, restaurants, and sightseeing points, are possible options to realize such an enhancement.

In summary, free-text responses indicated that factors such as route familiarity, perceived safety, scenic interest, and physical constraints influenced participants' decisions. These insights suggest that future navigation systems could benefit from incorporating contextual and affective factors, in addition to physical attributes like slope and distance.

## Discussion

This section discusses how our findings contribute to understanding user behavior in pedestrian navigation and what they imply for personalized route recommendation.

### Summary of key findings

Our results offer several insights across the three research questions (RQs):

- **RQ1**: Numeric gradient annotations modestly encouraged participants to select gentler, longer routes. The effect was more apparent in long-distance cases.
- **RQ2**: Graphical annotations (altitude profiles) did not significantly outperform numeric annotations. This may be due to insufficient visual differentiation or cognitive saturation.
- **RQ3**: Personality traits, especially those measured by SSS, significantly influenced route choices. Participants with higher scores in SSS subdimensions were more likely to prefer non-shortest, gentler routes.

These results suggest that although gradient information may not affect all users equally, it can influence preferences under certain conditions, particularly when detour costs are substantial.

**Interpreting SSS subtraits.** To interpret the SSS subdimensions, we draw on prior work in mobility and psychology. High ES scores reflect openness to exploration and align with greater acceptance of detours [18]. BS may indicate aversion to repetitive paths, encouraging deviation even at higher effort [37]. TAS relates to willingness to challenge norms, which may manifest as a readiness to avoid default shortest routes [17]. Dis reflects impulsiveness and low adherence to conventions, possibly supporting spontaneous route deviations. Together, these traits suggest that users employ differentiated cognitive strategies when evaluating route trade-offs, highlighting the potential for psychological profiles to inform adaptive navigation design.

**Implications and connections to prior work.** Our findings reinforce the value of incorporating user-specific traits into navigation systems. While multi-objective routing algorithms can generate diverse route sets [13,14], selecting appropriate alternatives requires behavioral data. Our study provides empirical support for using psychological profiles, particularly SSS scores, as a basis for personalized route recommendations.

Previous studies in vehicular routing and tourism have explored individual differences [16–18], but few have addressed how personality traits interact with spatial trade-offs in pedestrian settings. Our findings extend these efforts by offering a structured methodology for quantifying and modeling such interactions.

### Limitations and future directions

Several limitations merit consideration. First, the sample size–though comparable to related studies–was relatively small and limited to Japanese participants in urban settings. We note

that this study did not report formal effect sizes or confidence intervals for non-parametric comparisons. Future research may employ regression-based models or bootstrapping to estimate these quantities and evaluate the practical magnitude of observed effects. Second, the visual annotation formats were not systematically designed or evaluated based on formal cartographic principles. Third, the study was conducted online, and while map interfaces approximated realistic usage, real-world context (e.g., weather, terrain, time pressure) was not replicated.

Future work should conduct in-the-wild experiments across broader populations and geographic areas. Integrating additional route factors—such as safety, scenery, or sidewalk quality—could extend the scope of multi-objective evaluation.

Moreover, adaptive user interfaces that respond to psychological profiles in real-time could further personalize navigation systems. For example, future navigation systems could implement adaptive user interfaces that dynamically adjust route suggestions and visual presentations based on real-time estimations of user preferences or psychological profiles. By integrating experimentally derived personality scores with behavioral logs, usage patterns, or physiological signals, such systems may enhance the precision of real-time user modeling. In summary, our study provides a foundation for incorporating personality-aware logic into pedestrian route recommendation systems, highlighting new design and modeling opportunities.

## Conclusion

This study investigated how pedestrians make route choices when presented with alternatives that trade off gradient and distance. Using a multi-objective optimization framework, we generated route pairs and evaluated user preferences under different visualization conditions. Our results showed that numeric gradient annotations modestly encouraged deviation from the shortest path in long-distance scenarios, whereas graphical representations offered no additional benefit. Importantly, personality traits, particularly the subdimensions of SSS, were significantly associated with route preferences. These findings underscore the value of incorporating psychological profiles into pedestrian navigation systems. Overall, this work contributes to the development of personalized, user-aware navigation tools and suggests future directions for integrating behavioral modeling into algorithmic route planning, including the development of adaptive user interfaces that dynamically respond to individual psychological traits.

## Supporting information

**S1 Appendix. Method details.** Details of preprocessing and implementation to prepare route pairs (route A and route B) were provided.
(PDF)

**S2 Appendix. Questionnaire sheet.** The questionnaire sheet was attached.
(PDF)

**S3 File. Dataset and script.** The anonymized and cleaned route-choice data and scripts to generate figures were attached as a ZIP file.
(ZIP)

## Author contributions

**Conceptualization:** Keisuke Otaki.

**Data curation:** Keisuke Otaki.

**Formal analysis:** Keisuke Otaki, Takayoshi Yoshimura.

**Methodology:** Keisuke Otaki, Takayoshi Yoshimura.

**Project administration:** Takayoshi Yoshimura.

**Resources:** Keisuke Otaki.

**Supervision:** Takayoshi Yoshimura.

**Validation:** Keisuke Otaki.

**Visualization:** Keisuke Otaki, Takayoshi Yoshimura.

**Writing – original draft:** Keisuke Otaki.

**Writing – review & editing:** Keisuke Otaki, Takayoshi Yoshimura.

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
