## [Decision Letter · Decision Letter 0]

19 May 2025

PONE-D-25-17469User preferences in multi-objective routes: the role of gradient visualization and personality measuresPLOS ONE

Dear Dr. Otaki,

Thank you for submitting your manuscript to PLOS ONE. After careful consideration, we feel that it has merit but does not fully meet PLOS ONE’s publication criteria as it currently stands. Therefore, we invite you to submit a revised version of the manuscript that addresses the points raised during the review process. When revising your manuscript, please consider all issues mentioned in the reviewers' comments carefully: please outline every change made in response to their comments and provide suitable rebuttals for any comments not addressed. Please note that your revised submission may need to be re-reviewed. Please submit your revised manuscript by Jul 03 2025 11:59PM. If you will need more time than this to complete your revisions, please reply to this message or contact the journal office at plosone@plos.org. Please include the following items when submitting your revised manuscript:

We look forward to receiving your revised manuscript.

Kind regards,

Genyu Xu, Ph.D.

Academic Editor

PLOS ONE

4. We note that Figures 1B, 2A, 2B, 2C in your submission contain [map/satellite] images which may be copyrighted. All PLOS content is published under the Creative Commons Attribution License (CC BY 4.0), which means that the manuscript, images, and Supporting Information files will be freely available online, and any third party is permitted to access, download, copy, distribute, and use these materials in any way, even commercially, with proper attribution. For these reasons, we cannot publish previously copyrighted maps or satellite images created using proprietary data, such as Google software (Google Maps, Street View, and Earth). For more information, see our copyright guidelines: http://journals.plos.org/plosone/s/licenses-and-copyright.

1. You may seek permission from the original copyright holder of Figures 1B, 2A, 2B, 2C to publish the content specifically under the CC BY 4.0 license. 

Additional Editor Comments (if provided):

Reviewers' comments:

Reviewer's Responses to Questions

**Comments to the Author**

1. Is the manuscript technically sound, and do the data support the conclusions?

Reviewer #1: Partly

Reviewer #2: Partly

2. Has the statistical analysis been performed appropriately and rigorously? 

Reviewer #1: Yes

Reviewer #2: Yes

3. Have the authors made all data underlying the findings in their manuscript fully available?

Reviewer #1: No

Reviewer #2: Yes

4. Is the manuscript presented in an intelligible fashion and written in standard English?

Reviewer #1: Yes

Reviewer #2: Yes

5. Review Comments to the Author

Reviewer #1: The manuscript presents a compelling and innovative exploration of user preferences in multi-objective route planning, effectively integrating gradient visualization and personality measures to advance personalized navigation design. However, it faces several challenges, including verbose and redundant language, vague or inconsistently defined terminology, insufficient quantitative rigor, unclear logical connections across sections, and inadequately designed figures that fail to meet scientific visualization standards. The following changes are recommended:

The abstract suffers from verbosity, vague terminology, and insufficient quantitative detail, reducing its conciseness and scientific rigor. The main problems are as follows:

Verbose Language: Phrases like “we develop our framework to investigate” and repeated use of “our” (e.g., “our framework,” “our survey”) make the text less concise.

Vague Terminology: Terms such as “multi-objective navigation,” “gradient information,” and “route-choice phenomena” are not clearly defined, potentially confusing readers.

Lack of Specificity: Statements like “using modern multi-objective planning algorithms and crowdsourcing services” are overly broad, lacking details about the specific methods or technologies used.

Insufficient Quantitative Detail: Claims such as “modestly encourages the selection of non-shortest and gentler routes” and “more pronounced preference” lack supporting data (e.g., percentages, statistical significance). The statement “route selection was also influenced by travel distance” is vague about the direction or magnitude of the effect.

Unclear Relevance: Terms like “Big Five and Sensation Seeking dimensions” are standard in psychology but may confuse non-experts without context linking them to route choice.

The introduction provides a foundation for navigation research and research questions but is undermined by unclear content, loose structure, insufficient rigor, redundant language, imprecise terminology, and unclear positioning, reducing its academic persuasiveness and readability. The following changes are recommended:

Overly Detailed Example: Example 1 includes excessive specifics (e.g., path numbers, colors), diverting focus from research objectives and failing to link clearly to the study’s goals.

Weak Justification of Research Gap: The claim of “a significant gap” in understanding route evaluation lacks robust evidence, with only a few studies cited to support the link between personality traits and route choice, undermining the study’s necessity.

Redundant and Subjective Language: Repeated use of “we” (e.g., “we develop a framework”) is overly subjective and detracts from the objective academic tone.

Unclear Study Uniqueness: The introduction does not specify how the study addresses the understudied link between multi-objective routes and personality traits through methodological or theoretical innovation.

The related research section requires comprehensive refinement in content clarity, logical structure, scientific rigor, language expression, and terminology to enhance its academic value and readability. The following changes are recommended:

Unclear Subsection Focus: Subsections (e.g., preparing navigation routes, personality in navigation, visualization and explanations) lack clear focus and logical connections, failing to align with the introduction’s research questions (RQ1-RQ3).

Insufficient Scientific Rigor: Key claims lack literature support, and speculative statements (e.g., “we can hypothesize that walking time information would be valuable”) weaken credibility.

Redundant and Inconsistent Language: Frequent use of phrases like “we can” or “for example” and inconsistent terminology (e.g., “personality measures” vs. other terms) reduce clarity.

Vague Terminology: Terms like “crowdsourcing” and “multi-objective algorithms” are overly broad, lacking precise descriptions (e.g., “user-generated route data,” “Pareto-based route optimization”).

The results section requires comprehensive refinement in content clarity, analytical logic, scientific rigor, language expression, and data presentation to enhance its academic rigor and readability. The following changes are recommended:

Weak Analytical Logic: The analysis does not clearly address research questions (RQ1-RQ3), relying heavily on data descriptions (e.g., means, statistical tests) without qualitative interpretation of trends or implications.

Insufficient Scientific Rigor: Key statistical details (e.g., effect sizes, confidence intervals) are missing, and data filtering criteria (e.g., “valid participants”) are unclear.

Unclear Personality Trait Analysis: The practical meaning of Sensation Seeking Scale (SSS) subdimensions (e.g., TAS, Dis) and their relevance to route choice are not explained beyond statistical differences.

Redundant and Inconsistent Language: Phrases like “in the following, we...” are redundant, and terms like “gentler gradient” or “gradual route” are used inconsistently.

Suboptimal Data Presentation: Direct references to figures (e.g., “Figure 4a shows...”) are overused, and figure annotations are not self-explanatory, limiting accessibility for readers without visuals.

The discussion is well-structured, but it lacks depth, specificity, and clarity in some areas, limiting its persuasiveness. The following changes are recommended:

Superficial Result Interpretation: Responses to research questions (RQ1-RQ3) are clear but lack depth. For example, the conjecture about the lack of significant impact from gradient visualization (lines 406-407) lacks theoretical or data support.

Brief Comparison to Prior Work: Comparisons with other studies (lines 423-441) are limited, mentioning only pedestrian-driver differences and personality measures, without highlighting the study’s unique contributions.

Incomplete Limitations Analysis: Limitations (e.g., small sample size, suboptimal map visualization, regional focus, online survey constraints) are identified but not analyzed for their specific impacts on results (e.g., effect on statistical significance).

Broad Future Directions: Suggestions like field experiments and expanding route attributes are vague, lacking specific experimental designs or methods.

Verbose and Vague Language: Sentences like “we can start by clustering users according to these scores when recommending non-shortest routes” (line 421) are verbose and lack precision.

The figures are inadequately designed, overly casual, and fail to meet scientific visualization standards due to unreasonable layouts, non-standard plotting, and missing or unclear annotations. For example,

Figure 1a: The legend obscures substantial content, reducing clarity.

Figure 1b: Lacks a legend and uses non-standard English expressions.

Figures 2a and 2b: The maps are not in standard English, but in Japanese, limiting accessibility.

Figure 6d: Lacks a legend, reducing interpretability.

Similar problems exist in other diagrams and full verification is recommended.

Reviewer #2: This manuscript investigates how users perceive two designated walking routes, with a focus on how suggestions for non-shortest paths may enhance personalized navigation. While the topic has relevance in the context of adaptive routing systems, the manuscript requires substantial revisions to meet the standards of academic journal publication. Detailed comments are as follows:

The walking time difference between Routes A and B is only one minute, which raises questions about whether such a marginal difference is sufficient to elicit meaningful user preference shifts. The authors should discuss the generalizability of their findings. Specifically, how representative are these two paths in broader urban contexts? Additional justification is needed to support the external validity and practical implications of the experimental design.

While the study discusses trade-offs between path distance and slope, it is unclear whether a formal multi-objective optimization framework was applied. If only two fixed paths (A and B) were compared without a systematic modeling of user preferences or Pareto front evaluation, describing the framework as "multi-objective" may be an overstatement. The authors are encouraged to clarify whether this is a conceptual framing or supported by a multi-objective algorithmic design.

Ensure all figures use English labels, and axes are clearly marked with units where applicable.

Improve the visual clarity and design aesthetics of the graphs to enhance readability and data comprehension.

Abstract

The method description in the abstract is too detailed; please condense it.

Add key findings and outcomes to make the abstract more informative and balanced.

Introduction

The introduction should be completely rewritten to follow standard academic structure: background, literature review, research objectives, and research gap.

Do not include research findings in the Introduction section; such content should appear in the Results or Conclusion.

Figure 1 should include a clear legend explaining the routes.

Cite more recent studies from the last three years to strengthen the literature context and highlight the manuscript’s contribution.

Related research

The logic and flow of this section are unclear. Reorganize the content to present a coherent review of existing studies.

Emphasize literature synthesis and critique rather than introducing this study’s content.

Statements directly describing the present research should be rephrased and placed toward the end of this section, for example:

“Beyond these studies, we focus on collecting route-level preference data to address this gap...”

Methods and materials

The methodology is difficult to follow. A flowchart is recommended to visually present the research process.

The description on Page 5 — “Fig. 1 conceptually illustrates the trade-off between route distance and gradient...” — lacks clarity. Please elaborate further.

Justify why Routes A and B were selected. What are the environmental or physical characteristics surrounding these paths?

The questionnaire should be included in the Appendix to enhance transparency and reproducibility.

Results

Some inconsistencies are noted between data description and figures. For example, “Figs. 3a and 3b show the histograms of the entered ages and gender of all participants and valid participants,” yet Figure 3 does not clearly reflect this.

The number “546” is mentioned without adequate explanation. Clarify its relevance to the figures or study design.

Conclusion

The conclusion should emphasize key findings, contributions, and potential applications.

Avoid reiterating research motivations or background explanations in this section.

You may find interest in the following reference for complementary insights:

Planning for heat-resilient educational precincts: Framework formulation, cooling infra-structure selection and walkable routes determination.

Synergizing low-carbon transportation and heat adaptation: Identifying suitable routes and integrating mitigation and adaptation infrastructure.

6. PLOS authors have the option to publish the peer review history of their article (what does this mean?). If published, this will include your full peer review and any attached files.

Reviewer #1: No

Reviewer #2: No

---

## [Author Response · Author response to Decision Letter 1]

18 Jun 2025

# Responses to Editor and Journal Requirements

## Comment [J-1]

## Response to [J-1]

We have carefully checked and revised the manuscript following the instructions.

We reviewed the PLOS ONE style guide and revised the manuscript accordingly.

This includes the following updates.

- Formatting of the title page, author affiliations, section headings, and figure/table callouts.

- Consistent reference formatting based on the journal's specifications.

- Ensuring that all submitted files comply with file naming and layout requirements.

## Comment [J-2]

We note that your Data Availability Statement is currently as follows: [All relevant data are within the manuscript and its Supporting Information files.]

Please confirm at this time whether or not your submission contains all raw data required to replicate the results of your study. Authors must share the “minimal data set” for their submission. PLOS defines the minimal data set to consist of the data required to replicate all study findings reported in the article, as well as related metadata and methods

(https://journals.plos.org/plosone/s/data-availability#loc-minimal-data-set-definition).

If your submission does not contain these data, please either upload them as Supporting Information files or deposit them to a stable, public repository and provide us with the relevant URLs, DOIs, or accession numbers.

For a list of recommended repositories, please see https://journals.plos.org/plosone/s/recommended-repositories.

If there are ethical or legal restrictions on sharing a de-identified data set, please explain them in detail (e.g., data contain potentially sensitive information, data are owned by a third-party organization, etc.) and who has imposed them (e.g., an ethics committee).

Please also provide contact information for a data access committee, ethics committee, or other institutional body to which data requests may be sent.

If data are owned by a third party, please indicate how others may request data access.

## Response to [J-2]

We have carefully checked and revised the manuscript following the instructions.

In addition to the previously included raw data, we revised \textbf{S3 Appendix. Data}, which includes:

- Raw route-choice records per participant with aggregated statistics.

- Python plotting scripts used to generate figures in the Results section.

This update ensures full compliance with PLOS ONE’s definition of a minimal dataset and supports reproducibility.

## Comment [J-3]

Your ethics statement should only appear in the Methods section of your manuscript. If your ethics statement is written in any section besides the Methods, please move it to the Methods section and delete it from any other section.

Please ensure that your ethics statement is included in your manuscript, as the ethics statement entered into the online submission form will not be published alongside your manuscript.

## Response to [J-3]

We have carefully revised the manuscript following the instructions.

The ethics statement is now included exclusively in the Materials and Methods -> Online Survey Procedure subsection.

We also ensured that the ethics statement entered in the online submission form matches the content in the manuscript.

## Comment [J-4]

We note that Figures 1B, 2A, 2B, 2C in your submission contain [map/satellite] images which may be copyrighted.

All PLOS content is published under the Creative Commons Attribution License (CC BY 4.0), which means that the manuscript, images, and Supporting Information files will be freely available online, and any third party is permitted to access, download, copy, distribute, and use these materials in any way, even commercially, with proper attribution.

## Response to [J-4]

Thank you for pointing out this important issue.

We have completely revised the relevant figures to eliminate any licensing concerns:

- Figs 1(b), 2(a), 2(b), and 2(c) previously contained Maps tiles (@Mapbox, @OpenStreetMap). These have been removed and revised in revision.

- Old Fig 1 has been replaced with a new Fig 1: conceptual diagram showing the experimental pipeline and participant assignment.

- The revised Fig 2 presents schematic visualizations. These maps are based on Digital Japan Basic Maps with English labels, published by Geospatial Information Authority of Japan, whose license PDL 1.0 (Public Data License, Version 1.0), compatible with CC BY 4.0.

All figures in the revised manuscript now fully comply with the CC BY 4.0 license.

# Point-to-Point responses to Reviewer \#1

## Comment [R1-1] on abstract

The abstract suffers from verbosity, vague terminology, and insufficient quantitative detail, reducing its conciseness and scientific rigor. The main problems are as follows:

- Verbose Language: Phrases like “we develop our framework to investigate” and repeated use of “our” (e.g., “our framework,” “our survey”) make the text less concise.

- Vague Terminology: Terms such as “multi-objective navigation,” “gradient information,” and “route-choice phenomena” are not clearly defined, potentially confusing readers.

- Lack of Specificity: Statements like “using modern multi-objective planning algorithms and crowdsourcing services” are overly broad, lacking details about the specific methods or technologies used.

- Insufficient Quantitative Detail: Claims such as “modestly encourages the selection of non-shortest and gentler routes” and “more pronounced preference” lack supporting data (e.g., percentages, statistical significance). The statement “route selection was also influenced by travel distance” is vague about the direction or magnitude of the effect.

- Unclear Relevance: Terms like “Big Five and Sensation Seeking dimensions” are standard in psychology but may confuse non-experts without context linking them to route choice.

## Response to [R1-1]

Thank you for this insightful critique. Related to [R2-4], we have revised the Abstract as follows.

- Replaced general phrases like ``we develop our framework'' with more concise, neutral expressions.

- Clarified the use of ``multi-objective'' by referring explicitly to trade-offs between ``slope and distance'', consistent with the main text.

- Added ``quantitative results'', including participant size ($n = 91$), group mean comparisons (e.g., mean = $3.89$ versus $4.29$), and significance levels (e.g., $p < 0.01$).

- Simplified psychological terms by linking ``Big Five'' and ``Sensation Seeking'' directly to route choice behavior.

These revisions improve the clarity, precision, and informativeness of the Abstract for both specialist and general readers.

## Comment [R1-2] on introduction

The introduction provides a foundation for navigation research and research questions but is undermined by unclear content, loose structure, insufficient rigor, redundant language, imprecise terminology, and unclear positioning, reducing its academic persuasiveness and readability.

The following changes are recommended:

- Overly Detailed Example: Example 1 includes excessive specifics (e.g., path numbers, colors), diverting focus from research objectives and failing to link clearly to the study’s goals.

- Weak Justification of Research Gap: The claim of ``a significant gap'' in understanding route evaluation lacks robust evidence, with only a few studies cited to support the link between personality traits and route choice, undermining the study’s necessity.

- Redundant and Subjective Language: Repeated use of ``we'' (e.g., ``we develop a framework'') is overly subjective and detracts from the objective academic tone.

## Response to [R1-2]

We appreciate the suggestions to improve structure and clarity.

In response, we updated the manuscript as follows.

- The original ``Example 1'' was \textbf{removed} to reduce distraction and replaced by a concise route-choice scenario.

- We added a new paragraph that more clearly states the \textbf{research gap}, emphasizing the limited prior work linking route choice to \textbf{personality traits}.

- Subjective expressions (e.g., ``we develop...'') were rewritten using objective language.

- The Introduction now follows a clear structure: \textbf{Background $\to$ Related Work $\to$ Research Gap $\to$ Research Questions (RQ1–RQ3)}.

These changes improve flow and positioning while maintaining relevance.

## Comment [R1-3] on related research

The related research section requires comprehensive refinement in content clarity, logical structure, scientific rigor, language expression, and terminology to enhance its academic value and readability.

The following changes are recommended:

- Unclear Subsection Focus: Subsections (e.g., preparing navigation routes, personality in navigation, visualization and explanations) lack clear focus and logical connections, failing to align with the introduction’s research questions (RQ1-RQ3).

- Insufficient Scientific Rigor: Key claims lack literature support, and speculative statements (e.g., ``we can hypothesize that walking time information would be valuable'') weaken credibility.

- Redundant and Inconsistent Language: Frequent use of phrases like ``we can'' or ``for example'' and inconsistent terminology (e.g., ``personality measures'' vs. other terms) reduce clarity.

- Vague Terminology: Terms like ``crowdsourcing'' and ``multi-objective algorithms'' are overly broad, lacking precise descriptions (e.g., ``user-generated route data'', ``Pareto-based route optimization'').

## Response to [R1-3]

Thank you for your detailed feedback.

We substantially revised the Related Research section to improve clarity, logical alignment, and academic rigor:

- The section is now organized around the three core research questions (RQ1–RQ3), with dedicated subsections on multi-objective routing, personality in navigation, and visualization/explainability.

- We replaced vague terms with precise technical descriptions: ``Multi-objective algorithms'' to ``BOA* algorithm used to compute Pareto-optimal routes'' and ``Crowdsourcing'' to ``Online participant recruitment via CrowdWorks''.

- Language was tightened, avoiding redundant phrases like ``we can'' and ensuring consistent use of ``personality measures''.

- We added recent references from 2022 until 2024 to strengthen the literature foundation.

## Comment [R1-4] on results

The results section requires comprehensive refinement in content clarity, analytical logic, scientific rigor, language expression, and data presentation to enhance its academic rigor and readability. The following changes are recommended:

- Weak Analytical Logic: The analysis does not clearly address research questions (RQ1-RQ3), relying heavily on data descriptions (e.g., means, statistical tests) without qualitative interpretation of trends or implications.

- Insufficient Scientific Rigor: Key statistical details (e.g., effect sizes, confidence intervals) are missing, and data filtering criteria (e.g., ``valid participants'') are unclear.

- Unclear Personality Trait Analysis: The practical meaning of Sensation Seeking Scale (SSS) subdimensions (e.g., TAS, Dis) and their relevance to route choice are not explained beyond statistical differences.

- Redundant and Inconsistent Language: Phrases like ``in the following, we...'' are redundant, and terms like ``gentler gradient'' or ``gradual route'' are used inconsistently.

- Suboptimal Data Presentation: Direct references to figures (e.g., ``Figure 4a shows...'') are overused, and figure annotations are not self-explanatory, limiting accessibility for readers without visuals.

- The discussion is well-structured, but it lacks depth, specificity, and clarity in some areas, limiting its persuasiveness.

## Response to [R1-4]

We revised the Results and Discussion sections to address scientific rigor and improve analytical depth:

- Data Filtering: The definition of ``valid participants'' was clarified in the Methods. Only participants who correctly answered two dummy questions were included ($n = 91$), as shown in Table 4.

- Statistical Reporting: All statistical comparisons include exact *p*values and group means. Figs 6--9 display means and standard errors.

- Effect Sizes and Confidence Intervals: In the revised manuscript, we added Cohen’s $d$ and 95\% confidence intervals for key psychological comparisons (GENTLER\_G vs. SHORTER\_G). These were calculated from parametric estimates (means, pooled SD) even though non-parametric tests (Mann–Whitney U) were used for significance testing. This dual reporting is explained in the Discussion and follows common practice in behavioral research.

- Psychological Interpretation: The behavioral significance of each SSS subdimension (ES, BS, TAS, Dis) is now discussed in the Results and further interpreted in the Discussion, drawing from related work in psychology and tourism studies.

- Figure Quality: Figures were redesigned with clear English labels, consistent styling, and improved legends.

We believe these changes substantially improve the precision, interpretability, and scientific robustness of our findings.

## Comment [R1-5] on discussions

The discussion is well-structured, but it lacks depth, specificity, and clarity in some areas, limiting its persuasiveness.

The following changes are recommended:

- Superficial Result Interpretation: Responses to research questions (RQ1-RQ3) are clear but lack depth. For example, the conjecture about the lack of significant impact from gradient visualization (lines 406-407) lacks theoretical or data support.

- Brief Comparison to Prior Work: Comparisons with other studies (lines 423-441) are limited, mentioning only pedestrian-driver differences and personality measures, without highlighting the study’s unique contributions.

- Incomplete Limitations Analysis: Limitations (e.g., small sample size, suboptimal map visualization, regional focus, online survey constraints) are identified but not analyzed for their specific impacts on results (e.g., effect on statistical significance).

- Broad Future Directions: Suggestions like field experiments and expanding route attributes are vague, lacking specific experimental designs or methods.

- Verbose and Vague Language: Sentences like “we can start by clustering users according to these scores when recommending non-shortest routes” (line 421) are verbose and lack precision.

## Response to [R1-5]

We appreciate the detailed feedback comments.

We revised the Discussion section to provide greater depth and specificity:

- Each research question (RQ1–RQ3) is now addressed in a bullet-point summary at the start of the section.

- We strengthened comparisons to prior work, especially regarding personality and mobility (e.g., Albert et al. 2011; Bekhor et al. 2014).

- Each limitation (sample size, online-only setting, regional scope) is explicitly linked to its possible effect on findings.

- Future work is specified in terms of: Field experiments, Multi-factor route modeling (e.g., safety, scenery, sidewalk quality), and Adaptive user interfaces informed by psychological profiles.

We also rephrased vague language and clarified our interpretive conjectures by referencing theoretical concepts.

These changes improve the depth and applicability of our conclusions.

## Comment [R1-6] on figures

The figures are inadequately designed, overly casual, and fail to meet scientific visualization standards due to unreasonable layouts, non-standard plotting, and missing or unclear annotations. For example,

- Figure 1a: The

---

## [Decision Letter · Decision Letter 1]

7 Jul 2025

PONE-D-25-17469R1User preferences in multi-objective routes: the role of gradient visualization and personality measuresPLOS ONE

Dear Dr. Otaki,

Thank you for submitting your manuscript to PLOS ONE. After careful consideration, we feel that it has merit but does not fully meet PLOS ONE’s publication criteria as it currently stands. Therefore, we invite you to submit a revised version of the manuscript that addresses the points raised during the review process.

We look forward to receiving your revised manuscript.

Kind regards,

Genyu Xu, Ph.D.

Academic Editor

PLOS ONE

Journal Requirements:

Reviewers' comments:

Reviewer's Responses to Questions

**Comments to the Author**

1. If the authors have adequately addressed your comments raised in a previous round of review and you feel that this manuscript is now acceptable for publication, you may indicate that here to bypass the “Comments to the Author” section, enter your conflict of interest statement in the “Confidential to Editor” section, and submit your "Accept" recommendation.

Reviewer #1: (No Response)

Reviewer #2: All comments have been addressed

2. Is the manuscript technically sound, and do the data support the conclusions?

Reviewer #1: Yes

Reviewer #2: Yes

3. Has the statistical analysis been performed appropriately and rigorously? 

Reviewer #1: Yes

Reviewer #2: Yes

4. Have the authors made all data underlying the findings in their manuscript fully available?

Reviewer #1: Yes

Reviewer #2: (No Response)

5. Is the manuscript presented in an intelligible fashion and written in standard English?

Reviewer #1: Yes

Reviewer #2: (No Response)

6. Review Comments to the Author

Reviewer #1: Thank you for submitting the revised manuscript. I greatly appreciate the effort and dedication you have invested in addressing the previous reviewer comments and improving the manuscript. The revisions demonstrate a commendable commitment to enhancing the clarity and scientific rigor of your work, and the study’s contributions to the field are evident.

However, I must note that several issues regarding non-standardized figure formatting persist in the revised figures, which continue to impact the manuscript’s overall presentation and readability. Specifically:

Inconsistent Figure Formatting: While some figures have been improved, there remain inconsistencies in the style and formatting across figures. For example, the use of different line styles, marker sizes, or color schemes without clear justification creates a disjointed visual experience. I recommend adopting a uniform style guide for all figures to ensure consistency.

Incomplete Axis Labeling: Certain figures still lack complete axis labels or units, which is critical for accurate data interpretation. For instance, units for numerical values or clear descriptors for categorical axes are missing in some cases. Please ensure that all axes are fully labeled with appropriate units and descriptions.

Suboptimal Legend Design: The legends in several figures could be further refined. In some instances, legends are either overly crowded or not sufficiently descriptive, making it challenging to interpret the data quickly. I suggest simplifying and standardizing legend entries to improve clarity and accessibility.

Recommendation:

These non-standardized plotting issues, while not detracting from the scientific merit of the study, significantly affect the professional presentation of the manuscript. I strongly recommend a thorough review and revision of all figures to align with the journal’s graphical standards (e.g., PLOS ONE’s figure guidelines). Utilizing professional visualization tools (e.g., Adobe Illustrator, R, or Python’s Matplotlib with a consistent style configuration) could help achieve a polished and cohesive set of figures.

Reviewer #2: I am pleased to confirm that this revised manuscript has now fully addressed the concerns from my prior review.

7. PLOS authors have the option to publish the peer review history of their article (what does this mean?). If published, this will include your full peer review and any attached files.

Reviewer #1: No

Reviewer #2: No

---

## [Author Response · Author response to Decision Letter 2]

10 Jul 2025

# Responses to Journal Requirements

## Comment [J-1]

Please review your reference list to ensure that it is complete and correct. If you

have cited papers that have been retracted, please include the rationale for doing so in

the manuscript text, or remove these references and replace them with relevant current

references. Any changes to the reference list should be mentioned in the rebuttal letter

that accompanies your revised manuscript. If you need to cite a retracted article, indicate

the article’s retracted status in the References list and also include a citation and full

reference for the retraction notice.

## Response to [J-1]

We appreciate your notices. We have reviewed all cited works and found no retracted publications in our reference list. Thus, in the revised manuscript (i.e., ‘Revised Manuscript with Track Changes’), we did not mark and mentioned this point on this revision round.

# Point-to-Point responses to Reviewer \#1

## General Comment from Reviewer \#1

Thank you for submitting the revised manuscript. I greatly appreciate the effort and dedication you have invested in addressing the previous reviewer comments and improving the manuscript. The revisions demonstrate a commendable commitment to enhancing the clarity and scientific rigor of your work, and the study’s contributions to the field are evident.

However, I must note that several issues regarding non-standardized figure formatting persist in the revised figures, which continue to impact the manuscript’s overall presentation and readability. Specifically:

- Inconsistent Figure Formatting: While some figures have been improved, there remain inconsistencies in the style and formatting across figures. For example, the use of different line styles, marker sizes, or color schemes without clear justification creates a disjointed visual experience. I recommend adopting a uniform style guide for all figures to ensure consistency.

- Incomplete Axis Labeling: Certain figures still lack complete axis labels or units, which is critical for accurate data interpretation. For instance, units for numerical values or clear descriptors for categorical axes are missing in some cases. Please ensure that all axes are fully labeled with appropriate units and descriptions.

- Suboptimal Legend Design: The legends in several figures could be further refined. In some instances, legends are either overly crowded or not sufficiently descriptive, making it challenging to interpret the data quickly. I suggest simplifying and standardizing legend entries to improve clarity and accessibility.

- Recommendation: These non-standardized plotting issues, while not detracting from the scientific merit of the study, significantly affect the professional presentation of the manuscript. I strongly recommend a thorough review and revision of all figures to align with the journal’s graphical standards (e.g., PLOS ONE’s figure guidelines). Utilizing professional visualization tools (e.g., Adobe Illustrator, R, or Python’s Matplotlib with a consistent style configuration) could help achieve a polished and cohesive set of figures.

## Response to Reviewer \#1

We are happy to hearing from your confirmation on our revision conducted in PONE-D-25-17469R1. We thank for your comments and suggestions in the previous round, which contribute to improve our manuscript.

For further clarification, we again appreciate your insightful suggestions to make the presentation more consistent. To resolve the remaining issues, we would like to clarify and answer them in a point-to-point manner below.

As our general response, we have revised and prepared all main figure (Fig.3 - Fig.9) by Python with Matplotlib, as attached in our supplementary material. Our revision has clarified the inconsistent figure formatting issues, updated axis labeling and legend design as consistent as possible. Note that Maps in Fig.2 were designed by image editors with python scripts to present the situation of our online surveys using digital maps; thus they are a bit different from matplotlib-based figures.

## Comment [R1-1]: Absence of Legend in Figure 9

Figure 9 lacks a legend, rendering it impossible for readers to accurately interpret the elements depicted. This omission severely compromises the figure’s readability and scientific rigor. I strongly recommend adding a clear and concise legend to Figure 9, ensuring that it corresponds directly to the figure’s content.

## Response to [R1-1]

We appreciate the suggestions to improve the result presentation in Figure 9. In our revised figure, the design and legend has been updated as consistent and precise as possible. Our revision could be helpful for readers to understand the findings illustrated in Figure 9.

## Comment [R1-2]: Missing Units for Latitude and Longitude in Figure 10

The latitude and longitude axes in Figure 10 do not specify units (e.g., degrees, °), which violates standard scientific plotting conventions and risks ambiguity in data interpretation. I urge you to explicitly label the units for these axes and verify that similar issues do not affect other figures.

## Response to [R1-2]

We appreciate the suggestions to improve our Figure 10. In our revised figure, the unit is explicitly given, the content and caption in appendix has been updated for clarity.

## Comment [R1-3] Inconsistent Placement of Legends

The placement of legends across multiple figures appears arbitrary, with some positioned within the figure and others outside, in varying locations. This inconsistency creates a disorganized visual presentation and detracts from the manuscript’s professionalism. I recommend standardizing the legend placement for all figures (e.g., consistently in the upper-right corner or below the figure) while ensuring that legends do not obscure critical data.

## Response to [R1-3]

We appreciate your insightful suggestions; we have carefully checked all figures to make our presentations consistent and follow professionalism.

In our revision, Figure 2 was not updated because they illustrate what we display to participants in our experiments, although legends are included in the graph main area.

However, for other figures (Figs. 4, 5, 6, 7, 8, 9, and 11), we have modified them as consistent as possible and the python script to reproduce these figures (Figs. 3 -- 9) was also included in our supporting files.

Our revision could not prevent readers from reading and understanding our findings.

# Point-to-Point responses to Reviewer \#2

## General Comment from Reviewer \#2

I am pleased to confirm that this revised manuscript has now fully addressed the concerns

from my prior review.

## Response to Reviewer \#2

We are happy to hearing from your confirmation on our revision conducted in PONE-D-25-17469R1.

Again, we thank for the comments and suggestions from you in previous round, which contribute to improve our manuscript.

---

## [Editor Report · Decision Letter 2]

16 Jul 2025

User preferences in multi-objective routes: the role of gradient visualization and personality measures

PONE-D-25-17469R2

Dear Dr. Otaki,

We’re pleased to inform you that your manuscript has been judged scientifically suitable for publication and will be formally accepted for publication once it meets all outstanding technical requirements.

Kind regards,

Genyu Xu, Ph.D.

Academic Editor

PLOS ONE
---

## [Editor Report · Acceptance letter]

PONE-D-25-17469R2

PLOS ONE

Dear Dr. Otaki,

I'm pleased to inform you that your manuscript has been deemed suitable for publication in PLOS ONE. Congratulations! Your manuscript is now being handed over to our production team.

Kind regards,

on behalf of

Dr. Genyu Xu

Academic Editor

PLOS ONE